# Parthenolide and Its Soluble Analogues: Multitasking Compounds with Antitumor Properties

**DOI:** 10.3390/biomedicines10020514

**Published:** 2022-02-21

**Authors:** Daniela Carlisi, Marianna Lauricella, Antonella D’Anneo, Anna De Blasio, Adriana Celesia, Giovanni Pratelli, Antonietta Notaro, Giuseppe Calvaruso, Michela Giuliano, Sonia Emanuele

**Affiliations:** 1Department of Biomedicine, Neurosciences and Advanced Diagnostics (BIND), Biochemistry Building, University of Palermo, 90127 Palermo, Italy; marianna.lauricella@unipa.it (M.L.); adriana.celesia@unipa.it (A.C.); giovanni.pratelli@unipa.it (G.P.); 2Laboratory of Biochemistry, Department of Biological, Chemical and Pharmaceutical Sciences and Technologies (STEBICEF), University of Palermo, 90127 Palermo, Italy; antonella.danneo@unipa.it (A.D.); anna.deblasio@unipa.it (A.D.B.); antonietta.notaro@unipa.it (A.N.); giuseppe.calvaruso@unipa.it (G.C.); michela.giuliano@unipa.it (M.G.)

**Keywords:** parthenolide, DMAPT, apoptosis, cell death, cancer therapy, NF-κB, oxidative stress

## Abstract

Due to its chemical properties and multiple molecular effects on different tumor cell types, the sesquiterpene lactone parthenolide (PN) can be considered an effective drug with significant potential in cancer therapy. PN has been shown to induce either classic apoptosis or alternative caspase-independent forms of cell death in many tumor models. The therapeutical potential of PN has been increased by chemical design and synthesis of more soluble analogues including dimethylaminoparthenolide (DMAPT). This review focuses on the molecular mechanisms of both PN and analogues action in tumor models, highlighting their effects on gene expression, signal transduction and execution of different types of cell death. Recent findings indicate that these compounds not only inhibit prosurvival transcriptional factors such as NF-κB and STATs but can also determine the activation of specific death pathways, increasing intracellular reactive oxygen species (ROS) production and modifications of Bcl-2 family members. An intriguing property of these compounds is its specific targeting of cancer stem cells. The unusual actions of PN and its analogues make these agents good candidates for molecular targeted cancer therapy.

## 1. Introduction

Parthenolide (PN) is a sesquiterpene lactone extracted from the medical herb feverfew (*Tanacetum parthenium*) that has been shown to exert both anti-inflammatory [1,2,3] and antitumor effects [4,5,6]. The biological activity of PN is correlated with its chemical structure, which includes an α-methylene-γ-lactone ring and an epoxide (Figure 1A). These components can interact with nucleophilic sites of specific cellular proteins [7]. In particular, the unsaturated double bonds conjugated with a carbonyl group have been shown to react with free protein thiol groups, causing modulation of the redox state in the cells and induction of cell death [8].

The anti-inflammatory action of PN has been widely considered a consequence of its inhibitory effect on the transcription factors belonging to NF-κB family [9,10,11]. These factors are responsible for the regulation of several diverse cellular events, including cell proliferation, differentiation and survival, and their role in inflammation is well established. NF-κB, upon specific stimuli, can promote the expression of proinflammatory mediators including cytokines such as TNF-α, IL-1 and IL-6 [12], adhesion molecules (e.g., ICAM, E-selectin) and enzymes including the inducible forms of cyclooxygenase (COX-2) and nitric oxide synthase (iNOS). Therefore, they behave as a central regulator to innate immunity and inflammatory processes [13,14].

Clinical and experimental data indicate that the NF-κB signaling pathway is the cause of cachexia that develops in cancer patients [15]. Indeed, patients often develop a wasting syndrome characterized by systemic inflammation and involuntary loss of body mass that cannot be reversed by normal nutritional support [15]. NF-κB inhibitors could also be used as anticancer therapy to treat cachexia [16,17].

As well as NF-κB inhibition, PN has been shown to directly inhibit platelet phospholipase A2, 5-lipoxygenase and inducible cyclo-oxigenase [18,19], thus contributing to inflammation prevention. In addition, evidence has been provided that PN can also inhibit the release of the proinflammatory mediators nitric oxide, PGE(2) and TNF-α from macrophages and TNF-α, IL-2, IFN-γ and IL-4 from human peripheral blood mononuclear cells [1].

According to its antiphlogistic action, PN, obtained by leaves or infusions of feverfew, has been widely used in folk medical practices for treatment of fever, arthritis and migraine [20,21].

In addition to its anti-inflammatory properties, PN has been shown to exert potent antitumor properties. Several recent papers highlight the role of PN as an antitumor agent and describe its ability to induce programmed cell death in a wide variety of tumor cell lines [22,23,24].

To make PN more usable in therapy, a more soluble derivative has been synthesized: dimethylaminoparthenolide (DMAPT) (Figure 1B), also known as LC-1 [25]. When formulated as a fumarate salt, DMAPT displays a solubility in water more than 1000 times greater than PN [26,27].

Using the main data present in the literature to date, this comprehensive review critically summarizes the biological effects of PN and its analogues in tumor cells, providing an up-to-date overview of their action mechanisms and selective antitumor potential that make these naturally derived compounds promising agents for cancer therapy.

## 2. The Effect of Parthenolide on Gene Expression Profile

Both the anti-inflammatory and antitumor effects of PN have been correlated with its ability to influence the expression of specific genes through modulation of some transcription factors.

### 2.1. Parthenolide and Nuclear Factor Kappa B (NF-κB)

The factors belonging to the NF-κB family play an important role in the transcriptional regulation of genes involved in cell survival and immune response [14]. The most well characterized NF-κB components are the heterodimers constituted by p65 and p50 subunits. Activation of this complex, as well as some other NF-κB dimeric forms, occurs as a prosurvival response following different proapoptotic stimuli including cytotoxic drugs [28,29], ionizing radiation [30,31], TNF-α [32,33], oxidative stress [34,35].

In resting cells, the NF-κB factors are sequestered in the cytoplasm by binding to their regulatory effectors’ IkB proteins. Upon specific stimulation, IkB proteins become phosphorylated on their N-terminal serine residues. This modification targets IkB through ubiquitination and consequent degradation by the 26S proteasome. NF-κB is thus released and can translocate to the nucleus where it promotes gene transcription [36].

The phosphorylation of IkB inhibitory proteins is controlled by specific kinases known as IkB kinases (IKK). In particular, the catalytic subunits IKKα and IKKβ together with the regulatory subunit NEMO, associate to form the active IKK complex (IKC) which can then phosphorylate IkB [37,38].

The inhibitory action of PN on NF-κB has been known for many years. Several lines of evidence indicate that PN can exert its inhibitory action by targeting IKC components [5,39]. A direct in vitro binding of PN to IKKβ has been demonstrated through the alpha-methylene gamma-lactone moiety present in the molecule [40]. Moreover, Idris et al. have shown that PN prevents the activation of both IKKα and IKKβ and suppresses growth and migration of mammary carcinosarcoma cells [41]. Inhibition of IKK results in the accumulation of IkB inhibitory proteins with consequent sequestration of NF-κB in the cytoplasm.

In addition, it has been shown that PN can directly affect the translocation of the p65 NF-κB subunit to the nucleus [42,43].

It is well known that NF-κB factors can promote the transcriptional activation of prosurvival/antiapoptotic genes including Bcl-X_L_, survivin and c-FLIP [44,45]. Inhibition of NF-κB by PN thus results in a reduction in the level of these gene products with consequent apoptotic effects. Many authors have recently shown that the antitumor/proapoptotic effects of PN in tumor cells are, at least in part, due to NF-κB inhibition. Considering that NF-κB factors are involved in malignant transformation and tumor progression [46,47], the use of PN as NF-κB inhibitor can be particularly adequate for those tumor forms where increased activity or constitutive activation of NF-κB have been characterized.

Interestingly, NF-κB inhibition by PN has been shown to sensitize tumor cells to the effects of chemotherapeutic drugs [22,48,49] or radiation [50,51,52,53].

### 2.2. Parthenolide and STAT Transcription Factors

Other transcription factors which are targeted by PN belong to the signal transducer and activator of transcription (STAT) family. These proteins modulate a wide variety of biological processes including cell proliferation, differentiation and apoptosis as well as inflammation and immunity [54,55]. Activation of these proteins occurs following tyrosine-phosphorylation by members of the Janus Kinase (JAK) family, which are cytokine receptor associated kinases. For a complete description of STAT family members and JAK signal transduction, review by Awasthi et al. and Hu et al. provide updated insights [56,57].

Phosphorylated STATs can then dimerize and translocate to the nucleus, where they exert their transcriptional activity. STATs transcriptional targets include several antiapoptotic and prosurvival factors [58,59], thus indicating their oncogenic potential. In particular, among STAT family members, STAT-3 seems to play a key role in tumor transformation and malignancy [60,61]. Inhibition of STAT-3 in tumor cells leads to down-regulation of its transcriptional targets belonging to the Bcl-2 family including the antiapoptotic Bcl-2, Bcl-X_L_ and Mcl-1 [62]. Similar effects have been found in cutaneous T-cell lymphoma cells after treatment with a STAT-3 inhibitor [63]. Given the role of STAT proteins in contributing to cell survival, oncogenic activation of these factors, especially STAT-3, has been found in diverse types of tumors including hepatocellular carcinoma, colorectal cancer, breast cancer, bladder cancer and hematological malignancies [55,64].

Our previous studies performed in human hepatocellular carcinoma (HCC) cell lines, provided evidence that PN is able to sensitize TRAIL-induced apoptosis by reducing phosphorylated STAT-3 levels. This event appears to be related to the inhibitory effect exerted by PN on the activation of JAK proteins. Sensitization by PN to TRAIL stimulated the extrinsic mechanism of apoptosis in HCC cell lines with the activation of both caspases 8 and 3 [65].

PN has been shown to behave as an inhibitor of the JAK/STAT signaling pathway in response to cytokines of the IL-6 family [66]. Evidence has been provided that PN affects multiple steps of this pathway. For instance, the compound can either inhibit JAKs through binding to their SH groups, thus avoiding STAT phosphorylation [67] or directly block the DNA binding of STAT proteins in activated hepatic stellate cells [66].

In addition, recent evidence has been provided that PN can regulate crosstalk of toll-like receptor 4 (TLR4) with STAT-3, thus reducing the release of inflammatory cytokines inducing apoptosis in activated hepatic stellate cells [68].

It is interesting to note that STAT-3 inhibition by PN has also been correlated not only with apoptosis induction in tumor cells but also with reversed drug-resistance [69].

### 2.3. Parthenolide Interplay with Other Transcription Factors and Epigenetic Effects

Notably, the antitumor effect of PN has been correlated with reduced gene transcription of prosurvival mediators [70].

In this context, AP-1 is another important transcription factor which is mainly involved in cell survival and proliferation [71,72]. Interestingly, PN has been shown to inhibit AP-1 DNA binding and transcriptional activity induced by UVB in a skin cancer model, thus exerting a chemopreventive activity [73].

Microphthalmia-associated transcription factor (MITF) is strongly associated with regulation of proliferation, survival and senescence of melanoma cells [74,75]. Hartman et al. have recently provided evidence that PN efficiently decreases the MITF M isoform level in melanoma cells as a result of transcriptional regulation that results in melanoma cell senescence [73].

The Wnt/β-catenin pathway is well known to be involved in tumor cell proliferation [76,77]. PN was recently shown to inhibit Wnt signaling by decreasing the levels of the transcription factors TCF4/LEF1 without affecting β-catenin stability or subcellular localization [78]. However, other authors also showed that PN is capable of reducing the levels of β-catenin in myelomonocytic leukemic U937 cells [70].

These findings might represent a nice tool to inhibit cell proliferation of those tumor types bearing β-catenin oncogenic mutations or Wnt signaling hyperactivation.

In addition to inhibition of transcription factors with prosurvival/oncogenic action, PN has been also shown to exert a function in promoting apoptosis through the activation of specific death genes. One example is given by the effect of PN on the tumor suppressor/proapoptotic transcription factor p53. It is well known that p53 is key regulator in a molecular network establishing the cell fate due to its ability to promote cell cycle arrest and/or apoptosis [79,80,81]. The p53 pathway is often abrogated in cancer and sometimes this might depend on increased activity of its negative regulator MDM2 [82].

In this regard, Gopal et al. have reported that PN can stimulate the ubiquitination of MDM2 thereby activating p53 cellular functions [83]. More recently, activation or stabilization of p53 by PN in tumor cells has also been found by other authors [17,84,85].

However, other reports indicate that PN can also act independently of p53 [86] and promote epigenetic modifications. For instance, epigenetic modulation of the CDK inhibitor and tumor suppressor p21 by PN has been evidenced in cancer cells [87].

Through an integrated molecular profiling approach, a description of transcription factors modulated by PN was provided. PN was able to alter the binding of important transcription factors in prostate cancer including C/EBP-alpha, fos-related antigen-1 (FRA-1), HOXA-4, c-MYB, SNAIL, SP1, serum response factor (SRF), STAT3, X-box binding protein-1 (XBP1), and p53 [88]. PN interplay with some transcription factors and relative signaling is described in Figure 2.

Beyond direct modulation of transcription factors, PN has been shown to modulate gene expression through the controls of epigenetic events. Evidence has been provided, for instance, that PN is able to deplete histone deacetylase 1 protein (HDAC1) and to induce cell death through ataxia telangiectasia mutated (ATM) [89]. Liu et al. have shown that PN favors DNA hypomethylation through a dual effect on DNA methyltransferase 1 (DNMT1). These authors reported that the compound can either directly inhibit DNMT1 or repress its expression by targeting transcriptional factor Sp1 binding to the DNMT1 promoter [90]. Up-to-date and wide descriptions of epigenetic effects of PN are provided by the papers of Irshad et al. [91] and Freund et al. [2].

## 3. Parthenolide Can Specifically Modulate Signal Transduction Pathways

A number of papers have highlighted the ability of PN to modulate signal transduction components, thereby influencing cell response to various stimuli.

The role of PN in counteracting cytokine-mediated signaling through the inhibition of the JAK–STAT pathway has previously been noted [67]. Moreover, PN has been shown to inhibit IL-1- and TNFα-induced NF-κB activation [66]. Figure 3 summarizes the effects of PN on different signaling cascades.

### 3.1. Modulation of PKC and MAPKs by Parthenolide

In addition to its effect in counteracting proinflammatory cytokine-mediated signaling, PN has been shown to exert an influence on cascades involved in cell proliferation and differentiation.

Specifically, modulation of protein kinase C (PKC) and MAP kinase components (MAPKs) by PN has been described. For instance, papers published some time ago show the effects of PN on PKC. In particular, the activation of specific PKC isoforms by PN has been shown to increase the effects of ultraviolet B (UVB) on skin cancer [92]. Moreover, evidence has indicated that PN leads to activation of the PKC and MAPK pathways, thus enhancing the differentiation of leukemia cells induced by all trans retinoic acid [93].

Considering the complex and diverse role of PKC isoforms and MAPKs in the control of cell proliferation and apoptosis, it is not surprising that contradictory data can be found regarding the effect of PN on these mediators. In contrast to the findings of the previous authors, through a protein microarray analysis of PN action in prostate cancer, Kawasaki et al. have correlated the antiproliferative effect of the compound with its ability to downregulate prosurvival kinase effectors, including PKC isoforms, MAPKs, PI3 kinase (PI3-K) and CaM kinases [94].

More recently, in line with these findings, PN has been shown to potently inhibit the B-Raf/MAPK/Erk mitogenic pathway in tumor cells [95] and to target epidermal growth factor receptor and signaling either in vitro or in vivo in lung cancer models [96]. Evidence has also indicated that PN inhibits oncogenic focal adhesion kinase in breast cancer cells [97].

Moreover, in accordance with previous considerations regarding Wnt/beta-catenin signaling, direct interference by PN with this oncogenic pathway was evidenced in colorectal cancer [98].

Overall, it seems that targeting prosurvival/oncogenic signaling is among the modalities that account for the antitumor action of PN.

### 3.2. Effects of Parthenolide on Stress Kinases

The c-Jun N-terminal kinase (JNK) and the p38 kinase are members of the MAPK family that are involved in the response to cell stress and induction of apoptosis [99,100].

Many studies support a role of PN and its analogues in the activation of these kinases, an event that is often correlated with the induction of oxidative stress and the consequent execution of cell death. It has been demonstrated that the PN analogue DMAPT suppresses in vivo prostate cancer growth and induces apoptosis through reactive oxygen species (ROS) generation with subsequent JNK activation [27,43,101,102]. The same authors showed that interfering with JNK by shRNA reduces the antiproliferative and apoptotic effects of DMAPT [27]. More recently, in line with these findings, a novel PN analogue (CPPTL) displayed antineoplastic effects via the ROS/JNK pathway in acute myeloid leukemia [101]. However, other authors have previously shown that activation of JNK by PN can also occur independently of ROS generation and appears to be crucial to overcome resistance of breast cancer cells to TRAIL-induced apoptosis [102]. Similarly, Zhang et al. have demonstrated that PN can sensitize various human cancer cells to TNF-α-induced apoptosis and that this effect is counteracted by JNK-dominant negative overexpression or specific JNK inhibitor [43].

It has to be considered that the role of both JNK and p38 in apoptosis is rather controversial [103,104,105]. In contrast to the previous findings supporting an antiapoptotic role of JNK and p38, Won et al. have provided evidence that PN inhibits the activity of both the kinases, leading to the sensitization of JB6 cells to UVB-induced apoptosis [92]. In general, the exact role of JNK and p38 in cell death varies in relationship with a number of factors that include the cell type, the kind of stimulus and the crosstalk with other signaling pathways [106].

PN has also been shown to influence the apoptotic extrinsic pathway through the modulation of the signal cascade downstream death receptors such as Apo1/Fas and TRAIL.

In this regard, Qin et al. have found that PN is capable of reversing the suppression of Fas-mediated apoptosis by TNF-α in acute myeloid leukemia cells, an effect which is most likely due to NF-κB inhibition and the consequent repression of antiapoptotic genes [107].

Evidence has indicated that PN is able to overcome resistance to TRAIL-induced apoptosis. According to the findings of Nakshatri et al. [102] PN can induce sensitization to TRAIL in resistant breast cancer, an event that is correlated with JNK activity. The apoptotic effects of PN are specifically described in the next paragraph.

## 4. Parthenolide and Cell Death

### 4.1. Classic Apoptosis

Apoptosis represents the major form of programmed cell death in multicellular organisms, and its execution is prevalently dependent on the activation of caspases [108]. Upstream events that trigger caspase-dependent cell death include receptor-mediated apoptotic extrinsic pathway and mitochondria-mediated intrinsic pathway [109,110].

A number of papers have reported that PN induces apoptosis in tumor cells either stimulating the apoptotic extrinsic or intrinsic pathway (Figure 4) [111]. The involvement of caspases in PN-induced cell death has been widely documented [85]. In multiple myeloma, for instance, PN has been shown to be capable of inducing a rapid accumulation of cleaved products of caspase-8 and caspase-3 and, to a lesser extent, caspase-9. Consequently, PN induced caspase-dependent cleavage of the antiapoptotic factors MCL-1 and XIAP, thus promoting the execution of apoptosis [112]. However, the same authors suggested that PN cytotoxicity is likely partly caspase-dependent, as they found that pan-caspase inhibitor Z-VAD-fmk only partially protects the cells from the effect of PN.

Massive apoptosis was recently found following PN treatment in different both in vitro and in vivo tumor models [113].

Apoptotic activities of PN also account for synergistic action in combination with other antitumor compounds as described in the previous paragraph.

### 4.2. Caspase Independent Cell Death

It is widely recognized that the induction of nonapoptotic forms of programmed cell death (PCD) can represent a valid alternative when tumor cells appear to be resistant to classic apoptosis [114]. Caspase-independent PCD includes diverse forms of death that result from the activation of specific pathways and involve other proteases to trigger the terminal events that culminate in cell suicide.

The first evidence that PN could stimulate caspase-independent PCD with necrotic-like morphology (Figure 4) was provided by Pozarowski et al., who found that HL-60 cells died by necrosis, concurrent with atypical apoptosis, after exposure to PN [115]. These authors showed that the different effects induced by PN depended on the concentration used. They demonstrated the presence of activated caspase-3 in cells undergoing apoptosis after treatment with 10 µM PN. On the other hand, at 30 µM PN they observed that most cells did not show any evidence of caspase-3 activation and revealed a necrotic morphology. Necrotic-like features thus appeared at higher PN concentrations, although were not identified as real necrosis.

Nevertheless, necrotic-like phenotype induced by PN was not further characterized, and necroptosis, the main caspase independent necrosis-like PCD form, has not yet been documented to be specifically activated by the compound. However, we have previously found that PN is capable of increasing the expression of RIP1, a major kinase involved in necroptosis [116].

Among caspase-independent cell death types, the type that was originally called “apoptosis-like PCD” includes apoptosis inducing factor (AIF)-mediated cell death. This factor is released by mitochondria under specific stimuli and translocates to the nucleus, where it promotes chromatin condensation and cell death. It has been documented that PN is able to induce cell death through an AIF-dependent mechanism in melanoma and osteosarcoma cells. In this mechanism, dissipation of the mitochondrial transmembrane potential but not the activation of the caspases has been highlighted [117].

Supporting a possible switch toward PN-induced cell death without caspase involvement is the recent observation that, under particular circumstances, PN can reduce the expression of caspase 3 and caspase 9 and is also capable of suppressing their activation [118].

Considering these observations as well as strict crosstalk among different types of cell death, further studies are needed to clarify the mechanisms of PN-induced Caspase independent-PCD in tumor cells.

### 4.3. Autophagy

Autophagy is a physiological process that leads to the lysosomal degradation of cellular components and damaged organelles, representing a prosurvival cellular response to various types of stress [119,120]. Under certain conditions, this process can evolve into a death event, namely called autophagic cell death [121]. The activation of this pathway is fundamental for tumor cells that show resistance to classical apoptosis and can, therefore, induce recurrence [122,123].

PN can induce cell death through autophagy activation (Figure 4) without the involvement of caspases. Yang et al. [24] demonstrated that in human osteosarcoma cells, PN induces human osteosarcoma cell death by activating autophagy and mitophagy. PN induction of autophagic death is associated with increased ROS and does not involve caspases.

Activation of the autophagic mechanism by PN could be an excellent strategy to enhance the action of chemotherapeutic agents [124]. It has been shown that PN induces autophagy in human promyelocytic leukemia cell line HL-60 and in human epithelial carcinoma HeLa cell line. The researchers also highlighted that the autophagic mechanism is necessary for PN to activate apoptosis [125].

In glioblastoma (GBM) cells, the efficacy of chemotherapy is limited by intrinsic resistance [126]. PN can induce survival inhibition and trigger cell death through autophagic activation in the GBM U373 cell line [127].

## 5. Parthenolide and Oxidative Stress

Targeting intracellular redox pathways can be considered as a therapeutic approach for cancer [128]. Oxidative stress is a cell condition caused by increased production of reactive oxygen species (ROS) or reduced function of antioxidant defense systems [129]. It is well known that ROS can behave as mediators of apoptosis. Unbalanced intracellular redox status can, in fact, trigger specific events including alteration of mitochondrial function and activation of death signaling pathways [130].

Many lines of evidence indicate that PN can promote apoptosis in tumor cells through the induction of oxidative stress [131,132] (Figure 5). This was first observed by Wen et al., who found that PN-induced apoptosis in hepatoma cells was accompanied with depletion of glutathione (GSH), generation of ROS, reduction of mitochondrial transmembrane potential and activation of caspases. These effects were effectively abrogated by the antioxidant N-acetyl-l-cysteine (NAC) and enhanced by the GSH synthesis inhibitor buthionine sulfoximine (BSO) confirming the role of oxidative stress in PN-induced apoptosis [133].

The principal site of ROS production in the cell is represented by the mitochondrial respiratory chain [134]. It has been demonstrated that treatment of tumor cells with PN exerts profound effects on mitochondria through the involvement of proapoptotic members of the Bcl-2 family, such as Bid, Bax and Bak. Consequent modifications across the mitochondrial membrane, determined by the proapoptotic Bcl-2 family proteins, lead to the subsequent release of mitochondrial death effectors, including cytochrome c and Samc/Diablo [135].

Kim et al. [136] have further shown that Bax translocation to the mitochondria correlates with ROS production during PN-induced apoptosis in cholangiocarcinoma cells. In particular, these authors have found that Bcl-XL-mediated inhibition of Bax translocation can decrease PN-induced ROS generation and subsequently inhibits the reduction in mitochondrial membrane potential and the release of proapoptotic factors from the mitochondria [136].

In addition, it has been reported that PN-induced apoptosis in multiple myeloma (MM) cells involves oxidative stress and that cell sensitivity depends on catalase activity [137]. Interestingly, the same authors found a different expression of catalase in MM cells than in normal lymphocytes, thus explaining the different response to PN. The compound appeared to be ineffective in normal cells that displayed a higher level of catalase [138].

Other authors have also supported the evidence that PN preferentially acts on tumor cells promoting oxidative stress rather than in normal cells [139]. They showed that PN causes oxidative stress in prostate cancer cells but not in prostate epithelial cells. This effect seems to be dependent on a selective stimulation of NADPH oxidase (NOX) by PN in tumor cells with a consequent decrease in thioredoxin reduction and downregulation of FOXO3a targets antioxidant enzyme manganese superoxide dismutase (Mn-SOD) and catalase [139].

PN and DMAPT also induced oxidative stress through activation of NOX in breast cancer cells [116,140]. It was highlighted that after a short treatment (1–8 h) with PN in MDA-MB231, there was an increase in O2•− induced by the activation of NADPH oxidase. With a longer treatment time, PN also induced mitochondrial production of O2•−, likely blocking mitochondrial activities. The induction of oxidative stress by PN was the main cause of reduced cell viability of breast cancer cells [116,140]. Experimental evidence has suggested that NOX involvement is a consequence of PN-induced activation of the EGFR receptor [141].

According to the findings of Kurdi et al., in cardiac myocytes, PN generation of ROS can depend on either mitochondria or NADPH oxidase in relationship with the concentration of the compound used [142]. Specifically, PN generated superoxide anion, and at lower concentrations (<5 μM) the source of superoxide was mainly mitochondria; at higher concentrations (>5 μM) the primary source was NADPH oxidase [142].

Although the precise mechanism of PN induced oxidative stress remains unclear, it seems that its ability to bind thiol groups can account for its effect on reduction of intracellular GSH and protein thiols, such as thioredoxin [139]. In accordance with this hypothesis, a recent report has described the role of PN in modifying extracellular protein thiol groups and identified surface thioredoxin-1 as one of the targets of PN in lymphoma cells [8]. These authors, however, did not provide a precise explanation of how altering the redox state of exofacial thiols modulates cell death and speculated that surface thioredoxin can mediate the crosstalk between exofacial thiols and downstream intracellular events [8].

A direct action of PN on intracellular thiols, including both free GSH and protein thiols, has been previously described by Zhang et al. [7]. These authors found that colorectal cancer cells underwent apoptosis following treatment with PN through depletion of intracellular thiols and concomitant increase in ROS production and intracellular calcium [7].

It is evident that PN effects on the cell redox state and mitochondrial activity are key components of its action mechanism.

Interestingly, chemoresistance often correlates with an increase in antioxidant activity and a reduction in intracellular ROS level [134,143]. In fact, many cancer cells express high levels of the transcription factor Nrf2, which increases the antioxidant defenses of the cells [144,145]. PN inhibits chemoresistance by inhibiting the overexpression of both Nrf2 and downstream targets, such as MnSOD and catalase [146]. PN likely induces Nrf2 inhibition via increased expression of miR-29b-1-5p [147], a miRNA that regulates the inhibition of proliferation and invasion of triple-negative breast cancer cells (TNBC) [148].

## 6. Parthenolide Selectivity in Tumor Cells and Targeting Cancer Stem Cells

### 6.1. Selective Action of Parthenolide in Tumor Cells

Evidence has been provided that PN can selectively induce apoptosis in tumor cells rather than in the normal counterparts [86,112,137,139,149] thus displaying a high potential in cancer therapy.

In addition, in vivo studies have shown that PN is capable of suppressing tumor growth and, in some cases, preventing metastasis in different mice xenograft models including mammary carcinosarcoma [41], triple negative breast cancer [116,150], acute leukemia [26], prostate cancer [27,50,151] and renal carcinoma [152].

Interestingly, the antitumor efficacy of PN is also sustained by its ability to overcome cell resistance to conventional chemotherapeutic agents or other antitumor compounds.

The significant in vivo chemosensitizing properties of PN have been evidenced by Sweeney et al., who obtained reduction in metastasis and improved survival in a xenograft model of breast cancer following a combination treatment of PN with docetaxel [153].

Our previous studies demonstrated that PN and its soluble analogue DMAPT also have a selective effect against triple-negative breast cancer cells and not against normal cells (HMEC) [146]. Furthermore, it has been shown that the administration of DMAPT in nude mice carrying xenografts of MDA-MB231 cells results in inhibition of tumor growth, an increase in mouse survival and a reduction in the metastasis process [116].

To date, the only clinical trial based on PN administration to patients with cancer is reported by Curry et al. [154], who conducted a phase 1 trial to evaluate the pharmacokinetics and toxicity of the compound extracted by feverfew. PN was administered as a daily oral tablet in a 28-day cycle, and patients were evaluated for response after every two cycles. Based on their results, PN had no significant toxicity, but when administered in doses of up to 4 mg it did not provide detectable plasma concentrations. This limit was attributed by the authors to the fact that they could not provide desirable levels of PN with the feverfew preparation [154].

It should be noted that PN has no toxic effect in normal cells and can protect against oxidative stress. Some studies show that PN has scavenging activity against reactive oxygen species, thus protecting myoblast cells. In particular, it has been shown that PN modulates mitophagy induced by oxidative stress and protects C2C12 myoblasts from apoptosis, suggesting a potential protective effect against skeletal muscle diseases associated with oxidative stress [17].

### 6.2. Parthenolide and Cancer Stem Cells

Many studies have shown that in some forms of cancer, there are cells with specific stem-like properties called cancer stem cells (CSCs) [155]. CSCs are considered the “initiating” cells of the tumor and the main reasons behind tumor heterogeneity, resistance to therapy and recurrence [156,157].

Therefore, a drug capable of targeting both CSCs and cancer cells would be an excellent therapeutic strategy for the treatment of cancer patients. Several studies have shown that PN is a compound capable of acting on CSCs.

Breast cancer is the most common form of cancer diagnosed in women worldwide and, despite the development of new diagnostic and therapeutic methods, is a leading cause of death. The presence of CSCs is potentially the cause of treatment failure; drug resistance; metastasis; and relapse after surgery, chemotherapy and radiotherapy in breast cancer.

There are several studies showing the effect of PN in reducing the presence of CSCs in solid and hematological tumors [4,158,159,160], acting mainly on NF-κB inhibition.

The first studies on the cytotoxic effect of PN on CSCs focused on hematological tumors. Leukemia stem cells (LSCs) are the main cause of the onset, growth and recurrence of acute and chronic myeloid leukemia (AML and CML). PN induces apoptosis in primary human AML cells and CML cells, and in LSCs derived from AML without effect in normal hematopoietic cells. The apoptotic mechanism induced by PN is associated with the inhibition of NF-κB, the proapoptotic activation of p53 and the increase of reactive oxygen species (ROS) [26].

Recent studies have shown that PN is able to reduce the presence of LSCs in drug-resistant leukemia K562/ADM cells and increase their sensitivity to apoptosis induced by doxorubicin through the downregulation of P-gp mediated by NF-κB [161].

The cytotoxic effect of PN has also been highlighted in multiple myeloma cancer stem cells (MM-CSCs). Multiple myeloma (MM) is an incurable malignant plasma cell tumor in which nearly all patients succumb to relapse. In vitro studies conducted using a three-dimensional tissue culture system have shown that PN has a potent cytotoxic effect against MM-CSCs [162].

Triple-negative breast cancers (TNBCs) are aggressive forms of breast cancer associated with a high recurrence rate. Carlisi et al. [150] found that PN and its more soluble analog DMAPT induce cytotoxic effects in CSCs of TNBC. It has been shown that both PN and DMAPT reduce the formation of mammospheres in TNBC cell lines (MDA-MB231, BT20 and MDA-MB436). In particular, the compounds exerted a significant inhibitory effect on the viability of stem cells derived from the dissociation of mammosphere, inducing the generation of ROS, mitochondrial dysfunction and cell necrosis [150].

Zhou et al. [163] have demonstrated that PN can inhibit the formation of mammospheres in MCF-7, a breast cancer line responsive to estrogen and progesterone receptors. This effect is mediated by the inhibition of NF-κB activity.

PN was found to increase the cytotoxic effect in CSCs derived from breast tumors by vinorelbine, a semisynthetic vinca alkaloid that acts through disruption of microtubule assembly. Liu et al. [164] developed liposomes containing PN and vinorelbine. Their combined cytotoxic effect was evaluated in MCF-7 and MDA-MB-231 cells and in CSCs derived from them. The antitumor activity of the combined treatment of liposomes containing vinorelbine and PN was also evaluated on MCF-7 xenografts [164].

It is known that the 5-fluorouracil (5-FU) chemotherapeutic agent has a powerful cytotoxic effect on most cancer cells, but not on CSCs, which can make some cancers resistant to this drug. In general, resistance to chemotherapy is caused by the overexpression of one or more ABC transporters, including ABCG2 and ABCB1/multidrug resistance protein 1 (MDR1) [165]. Studies of nasopharyngeal carcinoma cells (NCS) have shown that while 5-FU treatment increases the CSCs population and COX-2 expression, PN reduces CSCs and downregulates COX-2. Studies conducted by Liao et al. [166] have shown that PN induces cell death in CSCs of NCS by inhibiting the NF-κB/COX-2 pathway.

## 7. Pathenolide Therapeutic Potential and Parthenolide Analogues

### 7.1. Synergistic Action of PN with Other Coumponds

From the evidence reported so far, PN represents a promising chemotherapy agent. Its effectiveness can be used in synergy with other anticancer agents of various types [22] (Table 1), with the aim of reducing the doses of the compounds used and preventing the onset of chemoresistance.

The combination of PN with Taxol has also been shown to be effective in lung cancer xenografts as described by Zang et al. [167]. Similarly, a PN analogue has been found to suppress in vivo prostate cancer growth and potentiate the effect of docetaxel [27]. The same authors have previously reported the in vivo effects of PN in combination with docetaxel in prostate cancer [168].

As described in the previous paragraph, significant results have also been found by Liu et al. [164], who combined PN stealthy liposomes with vinorelbine stealthy liposomes to treat breast cancer stem cells in both in vitro and in xenograft models.

In breast cancer, PN has also been shown to enhance sensitivity to antiestrogen such as tamoxifen [169] or fulvestrant [170], an effect which was mainly correlated with inhibition of NF-κB.

Other studies [171] of triple negative breast cancer cells have shown that PN sensitizes breast cancer cells to suberoylanilide hydroxamic acid (SAHA), a histone deacetylase inhibitor. One study showed that, in MDA-MB231 cells, the association of PN to SAHA inhibits the cytoprotective responses induced by the single compounds, while the cytotoxic effects are enhanced [171].

In lung cancer cells, Fang et al. have observed an increased susceptibility to low doses of oxaliplatin [172].

In addition to its anticancer properties, PN seems to exert beneficial effects in combination with chemotherapeutic drugs as reported by Francescato et al. [173] who showed its ability to reduce cisplatin-induced renal damage in mice.

More recently, evidence has indicated that PN strongly potentiates the effect of epirubicin in breast cancer cells [174].

Preclinical studies have also highlighted the efficacy of PN, or its analogues, in combination with nonconventional antitumor agents including COX-2 inhibitors [175], sulindac [176], steroid anti-inflammatory agents [177], retinoic acid [93], arsenic trioxide [178] and TRAIL [65,102].

Due to inhibition of NF-κB, PN has been shown to increase X-ray sensitivity of tumor cells [50,179]. Moreover, thermoenhancement effects of PN in lung adenocarcinoma cells have suggested a possible combination with hyperthermia as a novel approach in cancer treatment [180,181].

### 7.2. Parthenolide Anologues

It should be noted that the studies conducted with PN have highlighted both its potential and limitations. The limits of PN are due to its poor solubility in water and poor stability. Researchers have tried to overcome these limits to make the compound even more effective. Several studies have been conducted that have aimed to make PN more soluble and stable while preserving its bioactivities [2,158] (Table 2).

For example, researchers attempted to make PN soluble in water by adding a methyl group and formulating the DMAPT in the form of fumarate salt [26,27].

In a recent study by Jia X et al. [182], the structure of PN was combined with semicarbazone/thiosemicarbazone groups. Semicarbazones and thiosemicarbazones have been extensively studied for their anticancer activities. In fact, both chemical substituents can inhibit kinase activity involved in cell proliferation and the development of tumors. Furthermore, thiosemicarbazones are able to chelate transition metals with effective antitumor activity with the substituted semicarbazone. In Jia X et al’s study, 21 new derivatives of PN semicarbazone or thiosemicarbazone were synthesized. Most semicarbazone derivatives showed greater cytotoxicity against human tumor cell lines than PN. Five synthesized compounds, including four semicarbazones and one thiosemicarbazone, were tested in MC38 tumor-bearing mice. In vivo results showed that a specific semicarbazone greatly reduced tumor burden [182].

A valid strategy to increase the solubility of a drug and make it more selective towards cancer cells is nanoencapsulation. Based on these considerations, the researchers sought to improve the effect of PN.

Karmakar et al. [183] used carboxyl-functionalized nanographene (fGn) delivery to overcome the extreme hydrophobicity of PN. The authors found that administration of fGn augments the anticancer and apoptotic effects of PN in human pancreatic cancer Panc-1 cell lines. This effect was accompanied by an increase in ROS production and mitochondrial dysfunction [183].

Darwish NHE et al. [184] designed polylactide co-glycolide (PLGA) nanoparticles conjugated with antiCD44, a highly expressed transmembrane glycoprotein in leukemia cells and encapsulating PN, in order to improve the selectivity by recognizing tumor cells rather than normal cells. Studies conducted on leukemia cells Kasumi-1, KG-1a, and THP-1 have shown that PLGA-antiCD44-PN nanoparticles significantly reduce cell viability compared to PN alone [184].

Given the abundance of PN in nature by several researchers, it is used to semisynthesize other sesquiterpene lactones such as arglabin [185] (Figure 6A). Arglabin is a sesquiterpene lactone belonging to the guaianolide subclass, isolated from Artemisia species [185,186]. It has strong anticancer activities on leukemia and human oral squamous and lung cancer cells [187,188]. The antitumor action of arglabin is due to its ability to inhibit farnesyl transferase, an enzyme that causes the activation of the proto-oncogene RAS [186].

In addition, micheliolide (MCL), a sesquiterpene lactone that can be produced from PN [185] (Figure 6B), has shown great potential in antitumor treatment [189]. The antitumor action of MCL is due, as well as for PN, to the inhibitions of the NF-κB and the STAT3 signaling pathways [190,191,192]. Furthermore, MCL induces death in leukemia cells by irreversibly activating pyruvate kinase [193]. ACT001 (Figure 6C) is the fumarate salt of dimethylaminomicheliolide as an orally available derivative of MCL, synthesized from PN by Accendatech Co., Ltd. (Tianjin, China) [194]. ACT001 has a strong anticancer action in breast cancer cells [195,196]. It has been shown that ACT001 is also able to inhibit the proliferation of glioma stem cells (GSCs) [197]. Studies conducted in GSC xenografts have shown that ACT001 acts by inhibiting the adipocyte enhancer 1 binding protein (AEBP1), resulting in inhibition of AKT phosphorylation and cell proliferation [197].

## 8. Conclusions

PN exerts its antitumor action through different mechanisms including transcriptional regulation, epigenetic effects, signal transduction modulation and induction of oxidative stress. Through these modalities, PN is able to induce apoptosis or alternative forms of cell death including “necrosis-like” and “apoptosis-like” programmed cell death as well as autophagic cell death. The compound appears to display selectivity to tumor cells and a particular efficacy in cancer stem cells, which are the cause of relapses and resistance following chemotherapy. Many lines of evidence indicate that PN potentiates the effect of conventional chemotherapeutics or other antitumor drugs in different tumor models. Studies are still underway to make PN a more effective drug by increasing its stability and solubility. For this purpose, some PN analogues have been developed and are currently under investigation in different tumor models. Overall, PN and its analogues represent promising agents in cancer therapy, and studies on these compounds have served as a basis to synthesize or isolate new sesquiterpene lactones that are increasingly effective to specifically target cancer cells.

## Figures and Tables

**Figure 1 biomedicines-10-00514-f001:**
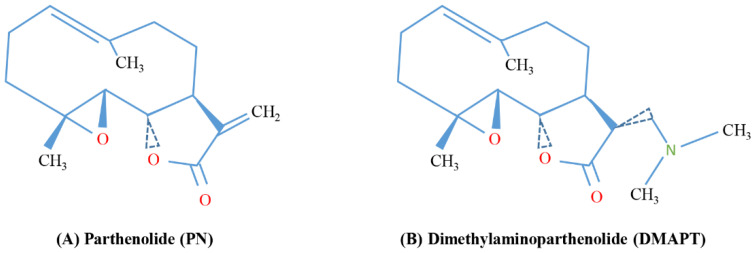
Chemical structures of (**A**) parthenolide (PN) and its analogue (**B**) dimethylaminoparthenolide (DMAPT).

**Figure 2 biomedicines-10-00514-f002:**
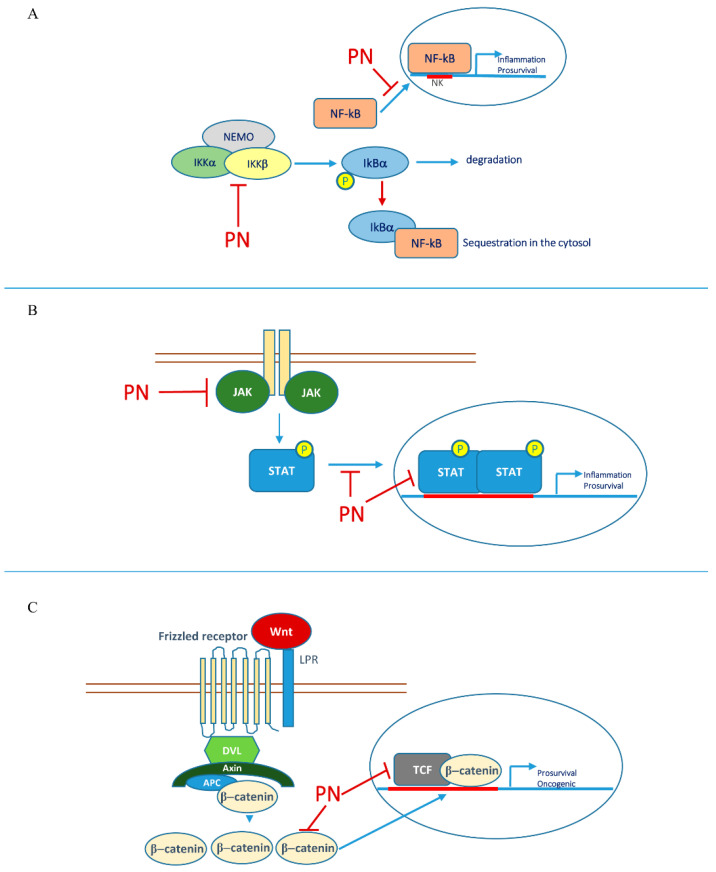
Effects of PN on prosurvival/oncogenic transcriptional factors and relative signaling: (**A**) NF-κB; (**B**) JAK/STAT; (**C**) β-catenin.

**Figure 3 biomedicines-10-00514-f003:**
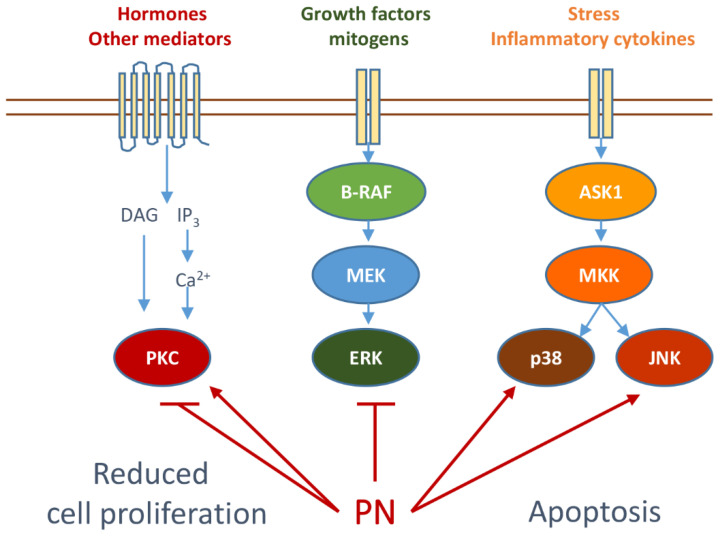
The effects of PN on different signaling cascades.

**Figure 4 biomedicines-10-00514-f004:**
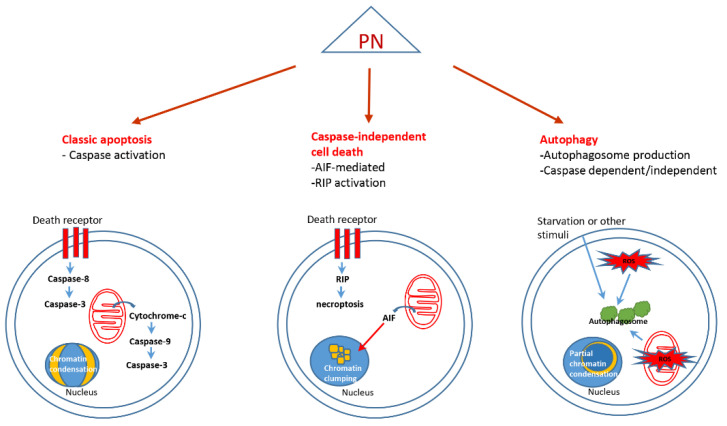
Different types of cell death induced by PN treatment.

**Figure 5 biomedicines-10-00514-f005:**
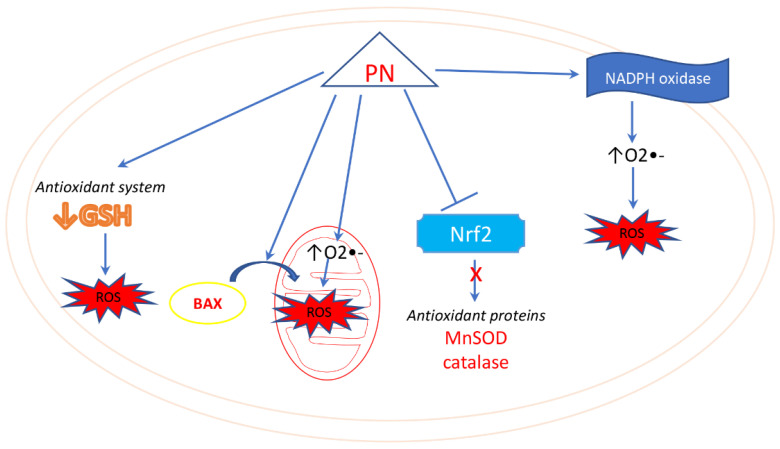
Mechanisms of oxidative stress induction by PN.

**Figure 6 biomedicines-10-00514-f006:**
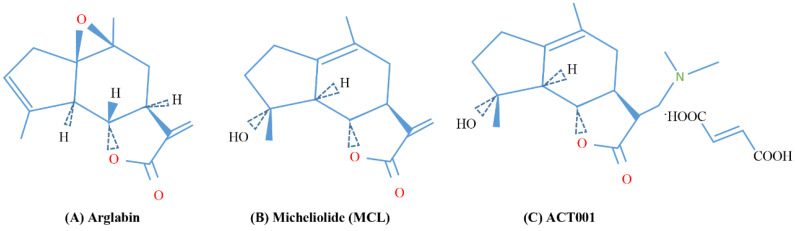
Chemical structures of (**A**) arglabin, (**B**) micheliolide (MCL) and its analogue (**C**) ACT001.

**Table 1 biomedicines-10-00514-t001:** Main classes of compounds that have a synergistic action in association with PN.

Synergistic Action of PN			
Coumpounds	Class of Compound	Tested on	Ref.
Taxol, Docetaxel, Vinorelbine	Antimicrotubule agents	Lung cancer, prostate cancer, breast cancer cell lines	[164,167,168]
Tamoxifen, Fulvestrant	Antiestrogen agents	Breast cancer cell lines	[169,170]
SAHA	Histone deacetylase inhibitor	Breast cancer cell lines	[171]
Oxaliplatin	Antineoplastic platinum drugs	Lung cancer cell lines	[172]
Epirubicin	Anthracycline doxorubicin analog	Breast cancer cell lines	[174]
NS398, sulindac	COX-2 inhibitors	Hepatocellular carcinoma cell lines	[175,176]
Dehydroepiandrosterone	Steroid anti-inflammatory agents	Pituitary tumor cell lines	[177]
Retinoic acid	Active metabolite of vitamin A	Leukaemia cell lines	[93]
Arsenic trioxide	Toxic metalloid	Leukaemia cell lines	[178]
TRAIL	Tumor necrosis factor family	Hepatocellular carcinoma and breast cancer cell lines	[50,65,179]
Radiotherapy	X-ray	Lung adenocarcinoma, prostate cancer cell lines	[50,179,180,181]

**Table 2 biomedicines-10-00514-t002:** Main compounds with structural and functional analogy to PN.

PN Anologues			
Coumpounds	Structure	Tested on	Ref.
Dimethylaminoparthenolide (DMAPT), also known as LC-1	PN with a methyl group also in the form of fumarate salt	Leukemia, prostate cancer, breast cancer	[26,27,116]
PN semicarbazone or thiosemicarbazone	PN with semicarbazone/thiosemi-carbazone groups	Colorectal carcinoma, glioblastoma, liver carcinoma, gastric cancer and lung cancer cell lines	[182]
PN-fGn or DMAPT-fGn	Carboxyl-functionalized nanographene	Pancreatic cancer cell lines	[183]
PLGA-anti CD44-PN nanoparticles	Polylactide coglycolide (PLGA) nanoparticles conjugated with antiCD44	Acute myeloid leukemia	[184]
Arglabin	Guaianolide sesquiterpene lactone	Leukemia, human oral squamous and lung cancer cell lines	[185,186,187,188]
Micheliolide (MCL)	Guaianolide sesquiterpene lactone	Hepatocellular carcinoma, leukemia cells	[185,189,190,191,192,193]
ACT001	Fumarate salt of dimethylaminomicheliolide	Breast cancer cells, glioma stem cells	[194,195,196,197]

## Data Availability

Not applicable.

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
