# Peer review of "Parthenolide and Its Soluble Analogues: Multitasking Compounds with Antitumor Properties"

_biomedicines, 2022, doi:10.3390/biomedicines10020514_

Round 1
Reviewer 1 Report
In this manuscript, the authors review the compound parthenolide, along with some analogs, and their potential to be used as cancer treatment therapies. Overall, I found the review to be well prepared, scientifically sound, thorough and informative. I recommend its publication. I found a couple minor spelling and grammar errors, that will easily be fixed during the editorial review prior to publication. I do wonder why only the structures of PN and DMAPT are shown, while several other compounds/analogs of PN are discussed.
Author Response
We thank the Reviewer for the positive consideration of our manuscript. We do appreciate the Reviewer’s opinion on the matter and recommendation for publication. English spelling and grammar has been revised.
Concerning the structures, we decided to show DMAPT together with PN since this analogue represents the main PN derivative and the most commonly used. In our opinion, including all the semicarbazone or thiosemicarbazone derivatives would have burdened the paper. However, we provided to include the structures of Arglabin, Micheliolide and ACT001 (new Fig 6).
All new or modified parts were inserted in a Tracked Change version in red.
Reviewer 2 Report
The authors provide the comprehensive review of reported activities of parthenolide. I admire the arduous work made by authors. The review covers majority studies on parthenolide and may be useful for those who interested with PN studies. Nevertheless, I have some comments that can improve the manuscript:
- There are repetitions that should be avoided, e.g. sentences in line 31, 81, 97. Please change or delete some of them.
- The information provided in subparagraphs 7.1 and 7.2 should be summarized in tables. That will help to use or compare them.
- Latin expressions should be typed in italic, please review entire manuscript.
- Last, but not least. I found the Conclusion section disturbing. It does not summarize the information provided, but exaggerate anti-cancer potential of PN. It is not advertisement section ;). Please rewrite.
Author Response
The authors provide the comprehensive review of reported activities of parthenolide. I admire the arduous work made by authors. The review covers majority studies on parthenolide and may be useful for those who interested with PN studies.
We thank the Reviewer for considering our manuscript complete and useful for the Journal readers.
We appreciate the positive Reviewer’s opinion and we are grateful for the suggestions that definitely improve the paper quality.
Nevertheless, I have some comments that can improve the manuscript:
- There are repetitions that should be avoided, e.g. sentences in line 31, 81, 97. Please change or delete some of them.
We thank the Reviewer for this observation. We have eliminated some repetition (line 81).
All new or modified parts were inserted in a Tracked Change version in red.
- The information provided in subparagraphs 7.1 and 7.2 should be summarized in tables. That will help to use or compare them.
Many thanks for this suggestion. We provided to include Table 1 that summarizes the main classes of compounds displaying synergistic action in association with PN and table 2 that summarizes the most representative PN analogues.
- Latin expressions should be typed in italic, please review entire manuscript.
We thank the Reviewer for this observation. We have modified as required.
- Last, but not least. I found the Conclusion section disturbing. It does not summarize the information provided, but exaggerate anti-cancer potential of PN. It is not advertisement section ;). Please rewrite.
Thanks for this suggestion. We modified the Conclusion section accordingly.
Reviewer 3 Report
The review "Parthenolide and its soluble analogues: multitasking com- 2
pounds with antitumor properties" is quite interesting and informative.
1) The authors should write the scientific names of species italicized, as for Tanacetum parthenium.
2) Check the configuration of the methyl group attached to the double bond in Parthenolide and Dimethylaminoparthenolide. It is not correct.
Is it a comprehensive review, or do the authors select a time range? In any case, this should be mentioned in the Introduction.
Author Response
The review "Parthenolide and its soluble analogues: multitasking com- 2
pounds with antitumor properties" is quite interesting and informative.
Many thanks for the positive evaluation
- The authors should write the scientific names of species italicized, asfor Tanacetum parthenium.
We thank the Reviewer for this observation. We have modified as required.
All new or modified parts were inserted in a Tracked Change version in red.
- Check the configuration of the methyl group attached to the double bond in Parthenolide and Dimethylaminoparthenolide. It is not correct.
Many thanks for this suggestion. We changed the structure of the compounds.
- Is it a comprehensive review, or do the authors select a time range? In any case, this should be mentioned in the Introduction.
We agree with the remark of the Reviewer.
It is a comprehensive review and we provided to specify it in the Introduction (line 69)